# Anticancer Activity of Amantadine and Evaluation of Its Interactions with Selected Cytostatics in Relation to Human Melanoma Cells

**DOI:** 10.3390/ijms23147653

**Published:** 2022-07-11

**Authors:** Danuta Krasowska, Agnieszka Gerkowicz, Paula Wróblewska-Łuczka, Aneta Grabarska, Katarzyna Załuska-Ogryzek, Dorota Krasowska, Jarogniew J. Łuszczki

**Affiliations:** 1Department of Dermatology, Venerology and Pediatric Dermatology, Medical University of Lublin, 20-090 Lublin, Poland; danuta.krasowska@umlub.pl (D.K.); agnieszka.gerkowicz@umlub.pl (A.G.); dorota.krasowska@umlub.pl (D.K.); 2Department of Pathophysiology, Medical University of Lublin, 20-090 Lublin, Poland; paula.wroblewska-luczka@umlub.pl (P.W.-Ł.); katarzyna.zaluska-ogryzek@umlub.pl (K.Z.-O.); 3Department of Biochemistry and Molecular Biology, Medical University of Lublin, 20-090 Lublin, Poland; aneta.grabarska@umlub.pl

**Keywords:** amantadine, melanoma, cisplatin, mitoxantrone, drug interaction

## Abstract

Patients with Parkinson’s disease are prone to a higher incidence of melanoma. Amantadine (an anti-Parkinson drug) possesses the antiproliferative potential that can be favorable when combined with other chemotherapeutics. Cisplatin (CDDP) and mitoxantrone (MTO) are drugs used in melanoma chemotherapy, but they have many side effects. (1) Clinical observations revealed a high incidence of malignant melanoma in patients with Parkinson’s disease. Amantadine as an anti-Parkinson drug alleviates symptoms of Parkinson’s disease and theoretically, it should have anti-melanoma properties. (2) To characterize the interaction profile for combinations of amantadine with CDDP and MTO in four human melanoma cell lines (A375, SK-MEL 28, FM55P and FM55M2), type I isobolographic analysis was used in the MTT test. (3) Amantadine produces the anti-proliferative effects in various melanoma cell lines. Flow cytometry analysis indicated that amantadine induced apoptosis and G1/S phase cell cycle arrest. Western blotting analysis showed that amantadine markedly decreased cyclin-D1 protein levels and increased p21 levels. Additionally, amantadine significantly increased the Bax/Bcl-2 ratio. The combined application of amantadine with CDDP at the fixed-ratio of 1:1 exerted an additive interaction in the four studied cell lines in the MTT test. In contrast, the combination of amantadine with MTO (ratio of 1:1) produced synergistic interaction in the FM55M2 cell line in the MTT (* *p* < 0.05). The combination of amantadine with MTO was also additive in the remaining tested cell lines (A375, FM55P and SK-MEL28) in the MTT test. (4) Amantadine combined with MTO exerted the most desirable synergistic interaction, as assessed isobolographically. Additionally, the exposure of melanoma cell lines to amantadine in combination with CDDP or MTO augmented the induction of apoptosis mediated by amantadine alone.

## 1. Introduction

Melanoma (malignant melanoma) is the most malignant skin cancer, characterized by dynamic growth and high mortality in humans. The incidence of melanoma is estimated at 4–6% per year worldwide, and it is constantly increasing [1]. Although melanoma is less common than other skin cancers, it accounts for the majority of deaths caused by skin cancer. If melanoma is diagnosed in its early stages, resection of the lesion is associated with a 90% 5-year survival rate [2]. However, advanced melanoma tends to metastasize and then has a poor prognosis.

Multidisciplinary experts have recently elaborated recommendations on treatment of melanoma, in which the first-line treatment of metastatic melanoma is target therapy or immune therapy [3]. The second-line treatment of melanoma is based on cytostatic drugs used in various combinations [3].

A recent clinical study indicates a higher incidence of melanoma in patients with Parkinson’s disease. It was noted that in this group, melanoma is less likely to metastasize and has a better prognosis compared to people without Parkinson’s disease [4]. It cannot be ruled out that the observed relationship is due to the protective effect of Parkinson’s disease and/or the use of anti-Parkinsonian drugs on melanoma development.

Among the drugs used in Parkinson’s disease, amantadine deserves special attention. First, it was used in early 2000s to treat the influenza A virus, but due to high resistance it was discontinued. Still, it used to treat some adverse effects of neuroleptic drugs and symptoms of Parkinson’s disease because it lessens bradykinesia, rigidity and tremor [5]. Amantadine inhibits N-methyl-D-aspartate (NMDA) receptors by accelerating the channel closing and stabilizing the closed state of the channel in NMDA receptors [6]. Amantadine is also an open-channel blocker of α4s2 nicotinic receptors [7] and α7 nicotinic receptors [8]. Amantadine inhibits calmodulin-dependent phosphodiesterase 1 (PDE1), and thus, increases adenosine-3′,5′-cyclic monophosphate (cAMP) concentration [9]. Amantadine blocks inwardly rectifying potassium channels (Kir2) but at drug concentrations ~3 times higher than therapeutically relevant doses [10]. Amantadine is also a potent ligand of sigma-1 receptors located intracellularly on the membranes of endoplasmatic reticulum and thus affects Ca^2+^ signaling [11]. Amantadine increased the expression of aromatic amino acid decarboxylase (AADC) [12]. In vitro experiments revealed that amantadine inhibits proliferation of HepG2 and SMMC-7721 cells by arresting the cell cycle at the G0/G1 phase [13]. Amantadine also reduces Bcl-2 and increases Bax proteins and mRNA levels in HepG2 and SMMC-7721 cell lines, which leads to the intensification of apoptosis in liver cancer cells [13]. Amantadine seems to be a drug with potential anti-cancer activity due to the fact that the drug could inhibit cell growth by modulating cyclin D1, cyclin E, CDK2 and inducing apoptosis via regulation of Bax and Bcl-2 proteins [13].

Disorders of apoptosis occur in melanoma and therapies targeting proteins from the Bcl-2 family may become an effective therapeutic option in melanoma malignant [14]. Cisplatin (CDDP) and mitoxantrone (MTO) are chemotherapeutic agents widely used in the treatment of several cancers. Platinum-based cisplatin shows antitumor activity against testicular, ovarian and bladder cancer but in combination with other drugs it can also be used to treat melanoma, breast, lung, colon, head and neck cancers [15]. Cisplatin binds irreversibly to DNA, causing cross linking of DNA, breaks in the DNA chain and missense mutations. Thus, it triggers apoptosis, particularly in rapidly dividing cancer cells [16]. In preclinical in vitro studies, CDDP plays a crucial role as a reference drug allowing comparison of anti-cancer effectiveness between drugs. MTO is an anthracenedione since it is an antineoplastic antibiotic derived from synthetic doxorubicin. Owing to its potent anticancer effect in humans, it is widely used clinically to treat acute leukemia, lymphoma, prostate cancer and breast cancer [17]. The MTO mechanisms of action include intercalation of DNA and inhibition of DNA and RNA synthesis [18]. Additionally, MTO is approved to treat different forms of multiple sclerosis as it presents immunosuppressive activity such as the inhibition of B-cell, T-cell and macrophage proliferation and a decrease in the secretion of tumor necrosis factor alpha (TNF-α) and interleukin-2 (IL-2) [19].

Management of melanoma, especially after metastasis, still remains a challenge. The treatment results of melanoma after immunotherapy or molecularly targeted therapies are still unsatisfactory and the 5-year survival rate reaches less than 30% [2]. Therefore, new drugs or their combinations are constantly being tested against melanoma [3]. The aim of this study was to assess the influence of amantadine on human melanoma cells, both primary and metastatic, and check the anticancer impact of amantadine in combination with well-known chemotherapeutics like MTO and CDDP [20,21,22].

## 2. Results

### 2.1. Cell Viability Test

Amantadine, CDDP and MTO inhibited the viability of human melanoma cell lines, both primary (FM55P, A375) and metastatic (FM55M2, SK-MEL28), in a concentration-dependent manner when applied separately.

Incubation of A375, SK-MEL28, FM55P and FM55M2 cells with increasing concentrations of amantadine resulted in the reduction of melanoma cell viability. The viability of the melanocytes (HEMa-LP) and keratinocyte cell line (HaCaT) was only slightly inhibited at high amantadine concentrations; however, the effect of this compound on cell viability at low concentrations was noticed. The significant inhibition of the viability of melanoma lines confirms the specific effect of amantadine on melanoma cells. Results are presented as mean ± SEM at each concentration (Figure 1).

### 2.2. Cell Proliferation Test

Amantadine administered alone dose-dependently reduced the proliferation of A375, SK-MEL28, FM55P and FM55M2 melanoma cells. The treatment of all the investigated cancer cells with increasing concentrations of amantadine reduced DNA synthesis that was evaluated by measuring BrdU (5-bromo-2′-deoxyuridine) incorporation into cellular DNA in proliferating cells (Figure 2).

### 2.3. Cytotoxicity Assessment—LDH Assay

The cytotoxicity of amantadine to normal human melanocytes (HEMa-LP), keratinocytes (HaCaT) and malignant melanoma cells (A375, SK-MEL28, FM55P and FM55M2) was tested by the LDH assay. The LDH enzyme is released into the culture medium as a result of damage to the cell membrane, which is a measure of irreversible cell death [23]. In our study, a significantly increased LDH leakage was observed in four amantadine-treated melanoma cell lines, while the effect on normal human HEMa-LP melanocytes and HaCaT keratinocytes was significantly lower and independent of amantadine concentration. The results are illustrated in Figure 3.

### 2.4. Flow Cytometry Analysis

Apoptosis is considered to be one of the main mechanisms of anti-cancer defense [24]. In order to determine whether the anti-proliferative effect of amantadine in melanoma cells was associated with apoptosis induction, we measured a population of cells with activated caspase-3 by flow cytometry. Caspase-3 is a well described protease and its activation is a hallmark of apoptosis [25]. In our studies, we found an increased number of caspase-3-positive cells versus control after treatment with amantadine at its IC_50_ concentration. It is noteworthy that the exposure of analyzed melanoma cell lines to amantadine in combination with CDDP or MTO augmented the induction of apoptosis mediated by amantadine alone (Figure 4).

The anti-viabilitive effect of amantadine was further examined by the analysis of cell cycle progression. The cell cycle profile was determined by staining the DNA with propidium iodide and measuring its intensity. Flow cytometry analysis identified accumulation of all studied melanoma cells in G1 phase after treatment with amantadine at its IC_50_ concentration. The final results are displayed as histogram peaks (Figure 5).

### 2.5. Western Blotting Analysis

To investigate the molecular mechanism of amantadine-mediated apoptosis and cell cycle arrest, the G1/S regulatory proteins and the related apoptosis-regulating proteins were examined. Western blotting analysis showed that amantadine at an IC_50_ concentration markedly decreased the cyclin-D1 protein levels and increased p21 levels. Additionally, amantadine significantly increased the Bax/Bcl-2 ratio (Figure 6).

### 2.6. Isobolographic Analysis of Interactions

Next, we analyzed the concentration-dependent inhibitory effects of CDDP and MTO on melanoma cells to find out if their combinations with amantadine could enhance the antitumor effects of these chemotherapeutic agents. The equations of log-probit concentration–response relationship curves (CRRC) for the studied drugs when used alone and for the combinations of CDDP and amantadine (Figure 7) and MTO and amantadine (Figure 8) allowed for the determination of the median inhibitory concentrations (IC_50_ values ± S.E.M.), which are summarized in Table 1. The IC_50_ values of amantadine for primary melanoma cell lines were lower than those for metastatic cell lines (Table 1).

The inhibitory effect of the tested compounds on melanoma cell viability was concentration-dependent with IC_50_ values ranging from 0.38 to 0.99 µg/mL (1.3 to 3.3 µM) for CDDP and from 0.016 to 0.901 µg/mL (0.03 to 1.7 µM) for MTO (Table 1). In the case of amantadine, the IC_50_ values for this drug ranged from 40.18 to 70.68 µg/mL (265.7 to 467.2µM) (Table 1). The test for parallelism between amantadine and CDDP revealed that the concentration–response relationship curves for amantadine and CDDP were mutually collateral for the cell lines FM55P and FM55M2 and non-parallel to each other for the cell lines A375 and SK-MEL28 (Figure 7). Similarly, the test for parallelism between amantadine and MTO revealed that the concentration–response relationship curves for amantadine and MTO were mutually collateral for the cell lines FM55P, FM55M2 and SK-MEL28 and non-parallel to each other for the cell line A375 (Figure 8).

Isobolographic analysis of interaction between amantadine and CDDP at the fixed-ratio of 1:1 showed addition in the studied melanoma cell lines (SK-MEL28, FM55P and FM55M2) and addition with tendency to synergism for the A375 cell line (Table 2 and Table 3; Figure 9).

The combination of amantadine and MTO at the fixed-ratio of 1:1 showed additive interactions for A375, SK-MEL28 and FM55P cell lines and synergy between amantadine and MTO (*p* < 0.05) for the FM55M2 cells (Table 4 and Table 5; Figure 10).

Additional confirmation of the additive interactions of the combination of amantadine with CDDP (Figure 11) and amantadine with MTO (Figure 12) are the results of the assessment of cell proliferation in the BrdU test.

## 3. Discussion

The relationship between Parkinson’s disease and melanoma has been a subject of research for many years, and the possible underlying connection is still not clear. Recent studies indicate a higher incidence of melanoma in patients with Parkinson’s disease, but in whom this tumor is less likely to metastasize and has a lower risk of death from this disease compared to patients without Parkinson’s disease. To the best of our knowledge, this is the first study to investigate the anticancer effect of amantadine on human melanoma cells, both primary and metastatic in vitro. The effect of amantadine on melanoma cells was compared to the effect of CDDP and MTO—two drugs used in the treatment of patients with melanoma. In our study, we observed that amantadine inhibits, in a concentration-dependent manner, cell viability in various melanoma cell lines, including primary and metastatic cell lines. Unfortunately, the anticancer activity of amantadine was observed at higher doses/concentrations of the drug than those reported for CDDP and MTO—two commonly used cytostatics in patients with melanoma. The anti-viabilitive effect of amantadine was determined by calculating the IC_50_ values, which ranged from 40.18 µg/mL (for FM55P) to 70.68 µg/mL (for SK-MEL28). The anticancer effect of CDDP for the same cell lines ranged between 0.54 µg/mL (for FM55P) and 0.99 µg/mL (for SK-MEL28). Similarly, the anti-viabilitive effect of MTO ranged between 0.183 µg/mL (for FM55P) and 0.901 µg/mL (for SK-MEL28). Results indicated clearly that the most resistant cell line to the selected treatment was SK-MEL28, whereas the FM55P cell line was more sensitive to the applied treatment.

Apoptosis is a form of programmed cell death that can be induced through the mitochondrial/intrinsic and death-receptor/extrinsic pathway. It has been shown that the mitochondrial pathway is regulated by B-cell Lymphoma 2 (Bcl-2) family proteins, including both pro-life (e.g., Bcl-2, Bcl-XL, Bcl-w) and pro-death (e.g., Bax, Bak, Bok) proteins [26]. When activated the Bax/Bak pro-apoptotic proteins undergo oligomerization at the mitochondrial outer membrane to mediate its permeabilization. Thereafter, the loss of the mitochondrial membrane potential’s irreversible release of cytochrome C and apoptosis-inducing factor (AIF) from the mitochondrial intermembrane space, and subsequently, the formation of the apoptosome are observed. Finally, the activation of executor caspase-3 and induction of apoptosis occur [27]. The cytotoxic effect of amantadine was related to apoptosis induction. Our results showed that the percentage of active caspase-3-positive cells were markedly increased after treatment with amantadine. It was associated with upregulation of the Bax/Bcl-2 ratio. Moreover, co-administration of amantadine with CDDP or MTO produced a pronounced increase in apoptotic cells. 

By means of the isobolographic analysis, we documented that amantadine additively interacted with CDDP and MTO in almost all the tested cell lines, except for one metastatic melanoma cell line (FM55M2), for which the combination of amantadine with MTO exerted synergistic interaction in the MTT test. A similar synergistic interaction in the MTT test has been observed for the combination of CDDP with osthole (a naturally occurring coumarin) in the FM55M2 cell line [28]. Although synergy in the anti-viabilitive effects of the tested drugs is the most desired combination (from a clinical viewpoint), the observed additivity between amantadine and CDDP or MTO is also favorable because the two-drug combinations offer the anti-cancer effect, and simultaneously, the drug doses/concentrations can be reduced. Considering the fact that amantadine in high doses can contribute to the elimination of melanoma cells, patients suffering from Parkinson’s disease, who additionally have confirmed melanoma, benefit twice when receiving amantadine because the drug can ameliorate the neurological status of the patients by minimizing Parkinson’s disease symptoms and can eliminate melanoma by negatively affecting melanoma cells. Noteworthy, amantadine did not affect normal human keratinocytes and melanocytes as it was confirmed in this study in both MTT and LDH tests.

It is not clear why the combination of amantadine with MTO synergistically inhibits viability in a melanoma cell line derived from a metastatic tumor (FM55M2) but not that from the primary tumor (FM55P). In this case, the additivity with a tendency towards synergy was observed for the combination of amantadine with MTO. Why the metastatic cells are more sensitive to the experimental treatment is not explained and it needs more extensive experiments. We can hypothetically accept that metastatic FM55M2 cells are more susceptible to the combined treatment with amantadine and MTO than primary FM55P cells. Perhaps, the diverse and different colony cells creating metastatic melanoma are sensitive to the studied drugs.

Explanation of this phenomenon, i.e., the selectivity of the combination of amantadine with MTO to produce synergistic interaction in terms of inhibition of cell growth in the metastatic cell line, seems to be a result of simple cooperation of the drugs in inhibition of cell growth, although a specific sensitivity of the metastatic cells to the drugs in combination cannot be excluded. Perhaps, both drugs (amantadine and MTO) affect various phases of the cell cycle, contributing finally to the cooperation of the two drugs in terms of inhibition of cell viability in various melanoma cell lines. Thus, many more malignant melanoma cells underwent induction of apoptosis than when the drugs were used separately. We can suggest that amantadine, through its various mechanisms, increases the sensitivity of the metastatic cells to MTO, resulting finally in increased cell death.

Another fact observed experimentally deserves more attention when extrapolating the results from this study to clinical conditions. In isobolography, doses and concentrations of the tested drugs in mixture are reduced in comparison to the doses and concentrations of the drugs when used alone. In case of the isobolographic synergy, the doses and concentrations of the drugs are substantially reduced, but the mixture produced the same effects as the effects observed for the drugs tested alone. Reduction of drug doses is clinically favorable due to the reduction of side effects produced by drugs used in high doses and concentrations. Lack or significant reduction of adverse effects, as a result of lowering drug concentrations, may become fully acceptable by patients (a favorable clinical outcome) when treated with high doses or concentrations of the drugs used separately.

## 4. Materials and Methods

### 4.1. Cell Lines

Primary (FM55P) and metastatic (FM55M2) malignant melanoma cells were purchased from European Collection of Cell Cultures (ECACC) and cultured in RPMI-1640 Medium (Sigma-Aldrich, St. Louis, MO, USA). Another two cell lines A375 (primary malignant melanoma) and SK-MEL28 (metastases malignant melanoma) were purchased from the American Type Culture Collection (ATCC) and cultured in Dulbecco’s Modified Eagle’s Medium—high glucose (DMEM) (Sigma-Aldrich, St. Louis, MO, USA) and Eagle’s minimal essential medium (EMEM), respectively. All culture media were supplemented with 10% Fetal Bovine Serum (FBS; Sigma-Aldrich, St. Louis, MO, USA) and 1% penicillin/streptomycin (Sigma-Aldrich, St. Louis, MO, USA). Cultures were kept at 37 °C in a humidified atmosphere of 95% air and 5% CO_2_. The cells grew to 80% confluence.

### 4.2. Cell Viability Assessment

A375, SK-MEL28, FM55P and FM55M2 cells were placed in 96-well plates (Nunc, Roskilde, Denmark) at a density of 1 × 10^4^ cells/mL, 3 × 10^4^ cells/mL, 2 × 10^4^ cells/mL and 2 × 10^4^ cells/mL, respectively. The next day, the culture medium was removed and cells were exposed to serial dilutions of amantadine, CDDP and MTO in fresh culture medium. Cell viability was assessed after 72 h by means of MTT method in which the yellow tetrazolium salt (MTT) is metabolized by viable cells to purple formazan crystals. Cells were incubated for 3 h with MTT solution (5 mg/mL, Sigma-Aldrich, St. Louis, MO, USA). Formazan crystals were solubilized overnight in sodium dodecyl sulfate (SDS) buffer (10% SDS in 0.01 N HCl), and the product was determined spectrophotometrically by measuring absorbance at 570 nm wavelength using microplate spectrophotometer (Ledetect 96, Labexim Products, Lengau, Austria). Each treatment was performed in triplicate and each experiment was repeated 3 times.

### 4.3. Cell Proliferation Assay—BrdU Assay

Cell Proliferation BrdU ELISA Kit (Roche Diagnostics, Mannheim, Germany) was used following manufacturer’s instructions. Optimized amounts of A375 (1 × 10^4^/mL) SK-MEL28 (3 × 10^4^/mL, FM55P (2 × 10^4^/mL), FM55M2 (2 × 10^4^/mL) cells were placed in 96-well plates (Nunc). On the next day, the cancer cells were treated for 48 h with increased concentrations of amantadine or mixtures of amantadine + CDDP or amantadine + MTO in fixed ratios of 1:1, followed by the addition of 10 µL/well BrdU Labeling Solution, and cells were then reincubated for additional 24 h at 37 °C. Then, BrdU assay were performed following manufacturer’s instructions. Quantitation was performed spectrophotometrically at 450 nm using microplate spectrophotometer (Ledetect 96, Labexim Products, Lengau, Austria).

### 4.4. Cytotoxicity Assessment—LDH Assay

Optimized amounts of A375 (1 × 10^4^/mL) SK-MEL28 (3 × 10^4^/mL, FM55P (2 × 10^4^/mL), FM55M2 (2 × 10^4^/mL),HaCaT (1 × 10^4^/mL) and HEMa-LP (5 × 10^3^/mL) cells were placed in 96-well plates (Nunc). The next day, cells were washed in PBS and then exposed to increasing concentrations of amantadine in fresh culture medium. The cytotoxicity was estimated based on the measurement of cytoplasmic lactate dehydrogenase (LDH) activity released from damaged cells after 72 h exposure to amantadine. LDH assay was performed according to manufacturer’s instruction (Cytotoxicity Detection KitPLUS LDH) (Roche, Mannheim, Germany). Briefly, 50 µL of cell medium was collected from each well, then 50 µL of Reaction mixture (freshly prepared) was added and incubated for 30 min at RT. Finally, 25 µL of Stop solution was added to each well on the 96-well plate. Absorbance was measured at two different wavelengths, one being the “measurement wavelength” (492 nm) and the other “reference wavelength” (690 nm), using microplate spectrophotometer (Ledetect 96, Labexim Products, Lengau, Austria). Maximum LDH release (positive control) was achieved by addition of Lysis buffer to untreated control cells. The average values of the culture medium background were subtracted from all values of experimental wells and the percentage of death cells was calculated in relation to the maximum LDH release [23].

### 4.5. Apoptosis Analysis

Optimized amounts of malignant melanoma cells were placed in six-well plates (Nunc). The next day, the medium was replaced with fresh medium containing amantadine or mixtures of amantadine + CDDP or amantadine + MTO at an IC_50_ concentration for 72 h. After that, cells were harvested and washed twice with PBS. Next, cells were fixed and permeabilized using the cytofix/cytoperm solution according to the manufacturer’s instructions of Phycoerythrin (PE) Active Caspase-3 Apoptosis Kit (BD Pharmingen). Finally, cells were washed twice in the perm/wash buffer prior to intracellular staining with PE-conjugated anti-active caspase-3 monoclonal rabbit antibodies. Labeled cells were analyzed by flow cytometer FACSCalibur (Becton Dickinson, San Jose, CA, USA), operating with CellQuest software to quantitatively assess the caspase-3 activity.

### 4.6. Cell Cycle Analysis

For cell cycle analysis, malignant melanoma cells were treated with amantadine at an IC_50_ dose for 72 h and then fixed/permeabilized in ice-cold 80% ethanol at −20 °C for 24 h. After fixation, the cells were stained with propidium iodide utilizing the PI/RNase Staining Buffer (BD Biosciences) according to the manufacturer’s instructions. The acquisition rate was at least 60 events per second in low acquisition mode, and at least 10,000 events were measured. Cell cycle analysis was performed by using a noncommercial flow cytometry analyzing software, Cylchred Version 1.0.2 for Windows (source: University of Wales) and WinMDI 2.9 for Windows (source: facs.scripps.edu/software.html, accessed 5 July 2022). The cells were acquired and gated by using dot plot FL-2 Width (X) versus FL-2 Area (Y)-gate to exclude aggregates, and analyzed in histograms displaying fluorescence two-area (yellow-orange fluorescence: 585 nm).

### 4.7. Protein Extraction and Western Blot Analysis

Optimized amounts of malignant *melanoma* cells were placed in 6-well plates (Nunc). On the following day, the culture medium was removed and the cells were treated with amantadine at an IC_50_ dose. After treatment, the cells were harvested, washed twice in ice-cold PBS, lysed in Radioimmunoprecipitation Assay (RIPA) buffer (Sigma) supplemented with protease inhibitor cocktail (Sigma), and centrifuged at 14,000× *g* for 10 min at 4 °C. The total protein concentrations were quantified spectrophotometrically using the Protein Assay Kit (Bio-Rad Laboratories, Hercules, CA, USA) according to the manufacturer’s instructions. For Western blot analysis, the protein extracts were solubilized in Laemmli buffer (0.19 M Tris-HCl, pH 6.8, containing 30% glycerol, 3% SDS, 0.015% bromophenol blue, and 3% β-mercaptoethanol) and boiled for 5 min at 100 °C. Then, equal amounts of the protein lysates were electrophoresed on SDS-PAGE and electroblotted to polyvinyl difluoride (PVDF) membranes (Merck Chemicals, Darmstadt, Germany). Next, the membranes were blocked in Tris Buffered Saline (TBS) with 5% non-fat dry milk and 0.05% Tween 20, pH 7.5, for 2 h at room temperature (RT) and incubated overnight at 4 °C with the following primary antibodies: cyclin D1, p21, Bax and Bcl-2 (Cell Signaling Technology, Beverly, MA, USA). Antibody dilutions were prepared according to the product data sheet. On the following day, the membranes were washed and then incubated with a horseradish peroxidase-labeled secondary antibody (Cell Signaling Technology Beverly, MA, USA) for 1 h at RT. Finally, the membrane was visualized using a Lumi-Light Western Blotting Substrate (Roche, Mannheim, Germany) according to the manufacturer’s instructions. Subsequently, stripping buffer (62.5 mM Tris-HCl, pH 6.8, with 100 mM β-mercaptoethanol and 2% SDS) was used to remove bound antibodies and the membranes were reprobed with anti-β-actin (Cell Signaling Technology, Beverly, MA, USA) used as a load control. Densitometric measurement of protein level was performed using ImageJ program. Obtained data were normalized to β-actin expression.

### 4.8. Isobolographic Analysis of Interactions

Log-probit analysis according to Litchfield and Wilcoxon [29] was used to determine the percentage of inhibition of cell viability per concentration of amantadine, cisplatin (CDDP) and mitoxantrone (MTO) when administered singly in the A375, SK-MEL 28, FM55P and FM55M2 cell lines measured in vitro by the MTT assay. Subsequently, from the log-probit concentration–response lines, median inhibitory concentrations (IC_50_ values) of amantadine and CDDP, MTO were calculated. The test for parallelism between two concentration–response curves (amantadine and CDDP, amantadine and MTO) was performed as described in our previous studies [28,30,31]. The median additive inhibitory concentrations (IC_50add_) for the mixture of amantadine with CDDP or MTO, which theoretically should inhibit 50% of cell viability, were calculated as demonstrated earlier [32,33]. The assessment of the experimentally-derived IC_50exp_ at the fixed-ratio of 1:1 was based on the concentration of the mixture of amantadine and CDDP or MTO that inhibited 50% of cell viability in cancer cell lines measured in vitro by the MTT assay. Details concerning the isobolographic analysis have been published elsewhere [34,35,36].

### 4.9. Statistical Analysis

GraphPad Prism 8.0 Statistic Software was used for statistical analysis of data. The calculations were performed by one-way analysis of variance (ANOVA test) for multiple comparisons followed by Tukey’s significance test. Data are expressed as the mean ± standard error (SEM) (* *p* < 0.05, ** *p* < 0.01, *** *p* < 0.001, **** *p* < 0.0001). The IC_50_ values for amantadine, CDDP and MTO, when administered alone and the IC_50exp_ values for the combinations at the fixed-ratio of 1:1 were calculated by computer-assisted log-probit analysis, as described elsewhere [28,29,30]. The experimentally-derived IC_50exp_ values for the mixture of amantadine with CDDP and amantadine with MTO were statistically compared with their respective theoretically additive IC_50add_ values by the use of unpaired Student’s *t*-test, as reported earlier [30,32].

## 5. Conclusions

The synergistic interaction determined isobolographically for the combination of amantadine with MTO in the human metastatic melanoma cell line (FM55M2) deserves further intensive investigation to reveal the molecular mechanism(s) of the action responsible for this favorable interaction. The additive interactions exerted by amantadine in combination with CDDP or MTO in various melanoma cell lines (A375, SK-MEL28, FM55P and FM55M2) can be also recommended for further in vitro experiments. If the results from this study extrapolate to clinical conditions, the anti-viabilitive effects of amantadine may be useful for patients suffering from melanoma.

## Figures and Tables

**Figure 1 ijms-23-07653-f001:**
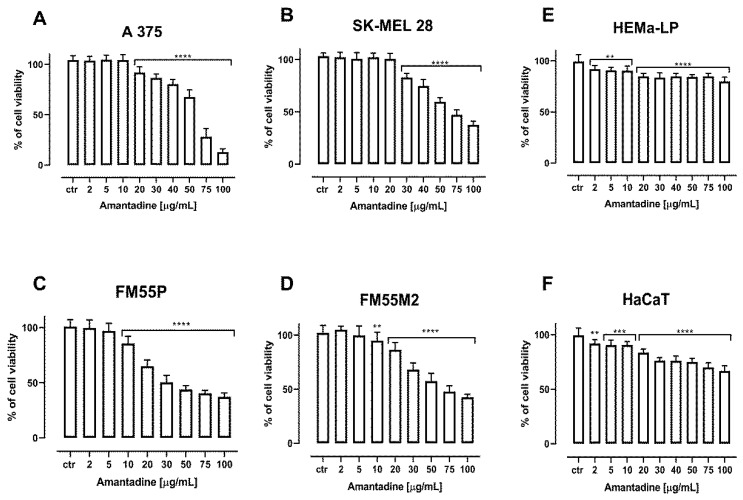
The anti-proliferative effect of amantadine administered against melanoma cell lines (A375 (**A**), SK-MEL28 (**B**), FM55P (**C**), FM55M2 (**D**)), melanocytes (HEMa-LP (**E**)) and keratinocyte (HaCaT (**F**)). Inhibition of cell viability was measured by the MTT assay after 72 h treatment with various increasing concentrations of the amantadine. Results are presented as mean ± SEM. (** *p* < 0.01; *** *p* < 0.001: **** *p* < 0.0001 vs. the respective ctr group).

**Figure 2 ijms-23-07653-f002:**
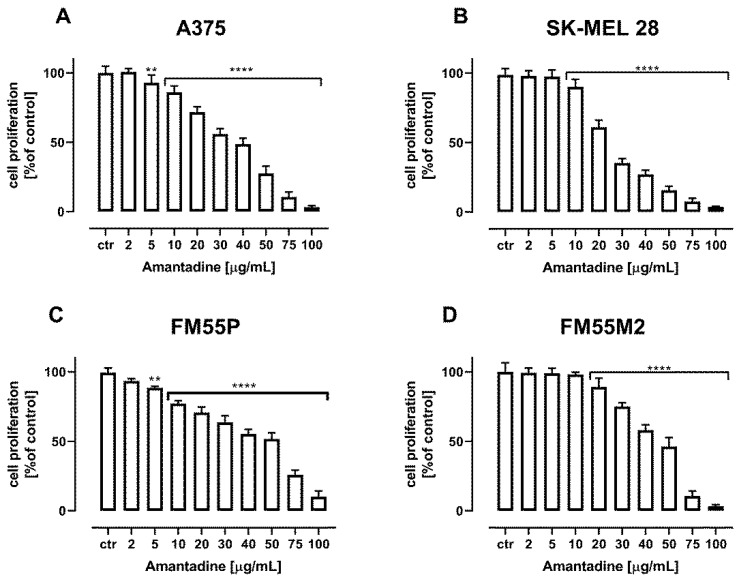
The effect of amantadine on the proliferation of malignant melanoma cell lines ((A375 (**A**), SK-MEL28 (**B**), FM55P (**C**) and FM55M2 (**D**)) was measured by BrdU assay after 72 h. Results are presented as mean ± SEM at each concentration (** *p* < 0.01; **** *p* < 0.0001).

**Figure 3 ijms-23-07653-f003:**
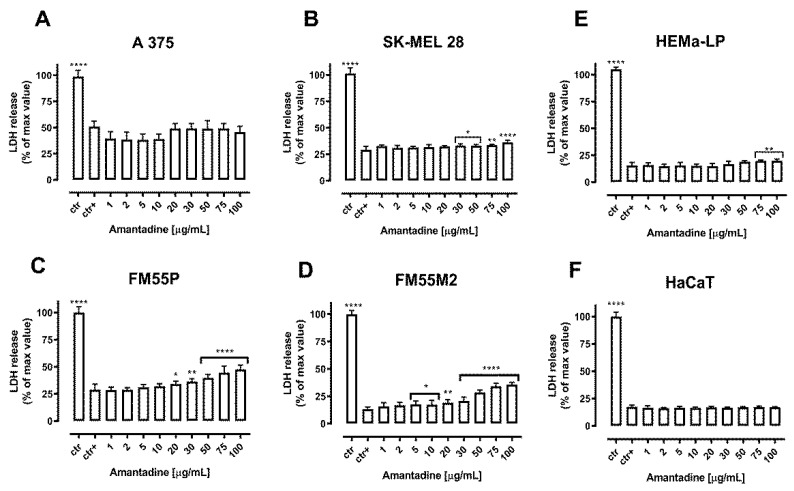
Cytotoxicity of amantadine to malignant melanoma cells (A375 (**A**), SK-MEL28 (**B**), FM55P (**C**), FM55M2 (**D**)), melanocytes (HEMa-LP (**E**)) and keratinocyte (HaCaT (**F**)) measured by LDH assay. Results are presented as the percentage in LDH release to the medium by treated cells versus cells grown in control medium (ctr+) and cells treated with Lysis buffer (ctr). Data are presented as mean ± S.E.M. * *p* < 0.05, ** *p* < 0.01, **** *p* < 0.0001.

**Figure 4 ijms-23-07653-f004:**
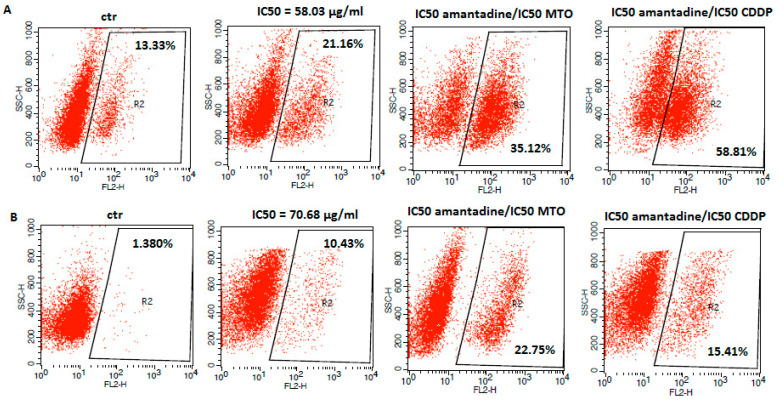
Representative flow cytometry dot plot graphs of A375 (**A**), SK-MEL28 (**B**), FM55P (**C**) and FM55M2 (**D**) melanoma cell lines after the treatment with a medium (ctr) and amantadine alone or in combination with CDDP and MTO at their IC_50_ concentrations for 72 h. Region R2 included apoptotic cells with active caspase-3.

**Figure 5 ijms-23-07653-f005:**
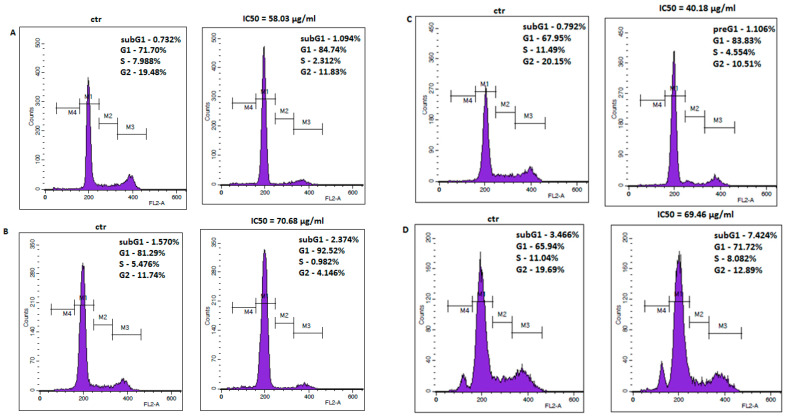
Representative flow cytometry histogram peaks of the A375 (**A**), SK-MEL28 (**B**), FM55P (**C**) and FM55M2 (**D**) melanoma cell lines after the treatment with a medium (ctr) and amantadine at an IC_50_ concentration for 72 h. Region M1, M2, M3 and M4 included G1, S, G2 and subG1 phase, respectively.

**Figure 6 ijms-23-07653-f006:**
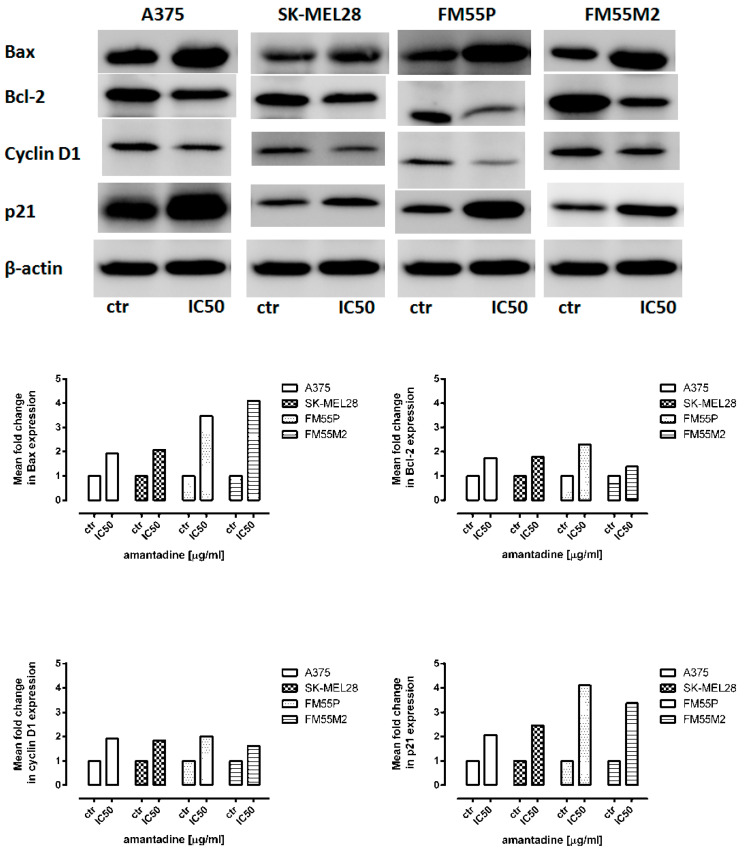
Representative blots of Bax, Bcl-2, cyclin-D1 and p21 proteins following incubation of melanoma cells with either culture medium alone (control) or amantadine at an IC_50_ concentration. β-actin was used as a gel loading control. Densitometric analysis of the bands was performed by ImageJ program.

**Figure 7 ijms-23-07653-f007:**
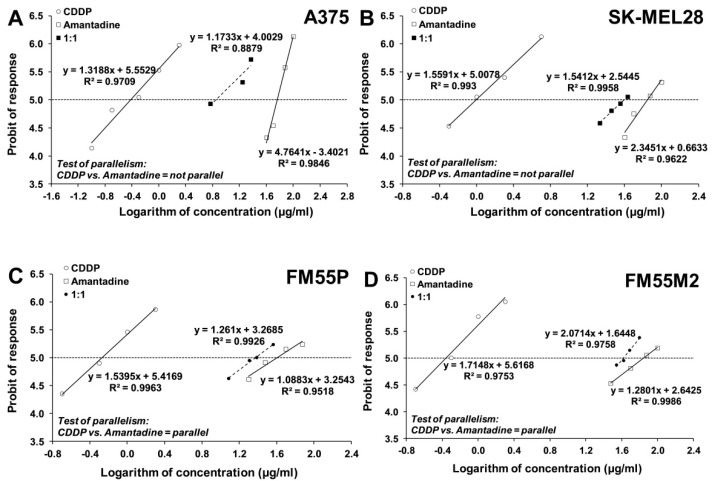
Concentration–response relationship curves (CRRCs) for cisplatin (CDDP) and amantadine administered alone and in combination at a fixed-ratio of 1:1, illustrating the anti-proliferative effects of the drugs in the human malignant melanoma cell lines (A375 (**A**), SK-MEL28 (**B**), FM55P (**C**) and FM55M2 (**D**)) measured in vitro by the MTT assay.

**Figure 8 ijms-23-07653-f008:**
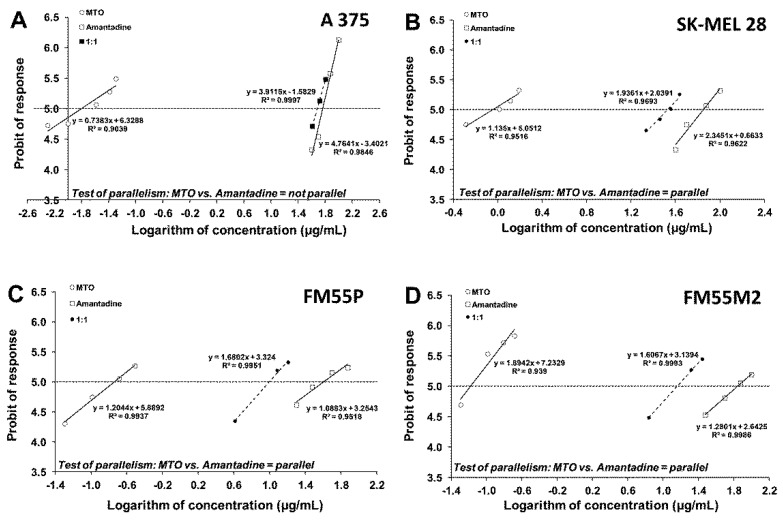
Concentration–response relationship curves (CRRCs) for mitoxantrone (MTO) and amantadine administered alone and in the combination at a fixed-ratio of 1:1, illustrating the anti-proliferative effects of the drugs in the human malignant melanoma cell lines (A375 (**A**), SK-MEL28 (**B**), FM55P (**C**) and FM55M2 (**D**)) measured in vitro by the MTT assay.

**Figure 9 ijms-23-07653-f009:**
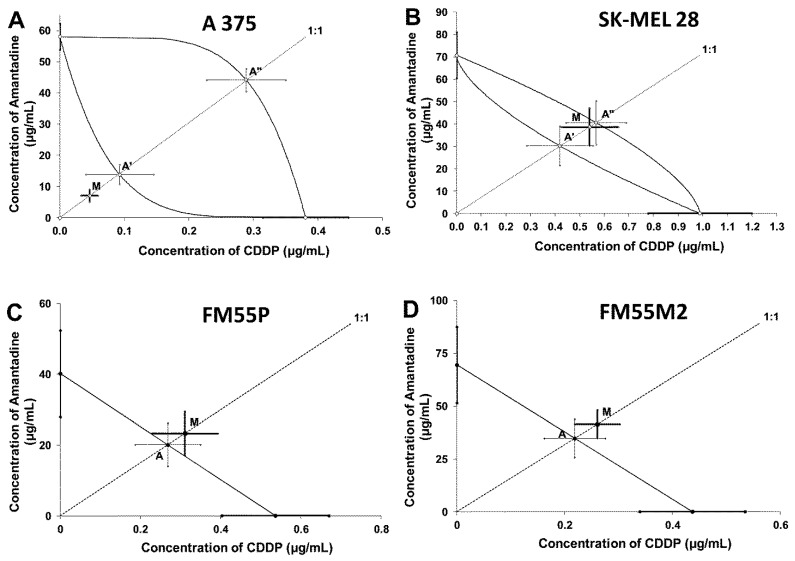
Isobolograms display additive interaction between amantadine and CDDP with respect to their anti-proliferative effects on A375 (**A**), SK-MEL28 (**B**), FM55P (**C**) and FM55M2 (**D**) melanoma cell lines measured in vitro by the MTT assay. The IC_50_ ± S.E.M. (as the error bars) for amantadine and CDDP are plotted on the X- and Y-axes, respectively. The points A, A′ and A″ depict the theoretically calculated IC_50add_ values (±S.E.M.). The point M on each graph represents the experimentally-derived IC_50exp_ value (±S.E.M.) for the mixture, which produced a 50% anti-viabilitive effect in the melanoma cell lines. Although point M is placed below the point A″ for the A357 cell line (**A**), the two-drug combination produced additivity in the MTT assay. For the SK-MEL28 cell line, the point M is placed between both points A′ and A″ within the area of additivity indicating additive interaction between amantadine and CDDP (**B**). In the case of the FM55P (**C**) and FM55M2 (**D**) cell lines, point M is placed slightly above point A, indicating additive interaction for amantadine with CDDP in the MTT assay.

**Figure 10 ijms-23-07653-f010:**
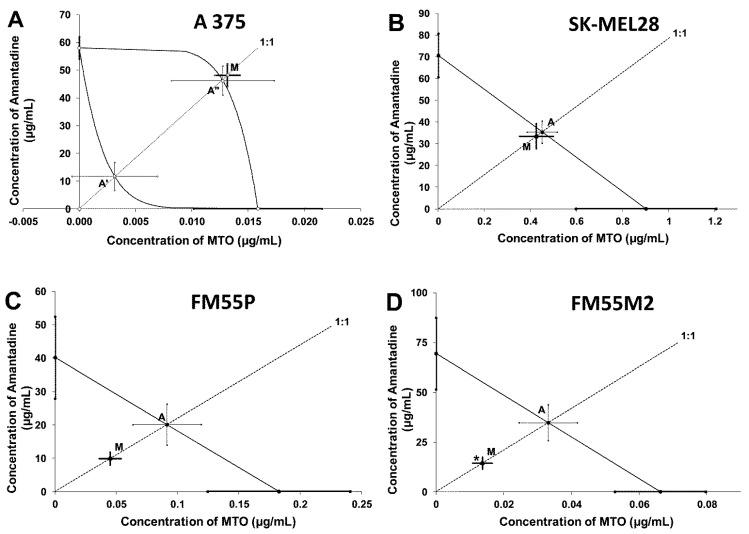
Isobolograms display additive and synergistic interactions between amantadine and mitoxantrone (MTO) with respect to their anti-proliferative effects on A375 (**A**), SK-MEL28 (**B**), FM55P (**C**) and FM55M2 (**D**) melanoma cell lines measured in vitro by the MTT assay. The IC_50_ ± S.E.M. for amantadine and MTO are plotted on the X- and Y-axes, respectively. The points A, A′ and A″ depict the theoretically calculated IC_50add_ values (±S.E.M.). The point M on each graph represents the experimentally-derived IC_50exp_ value (±S.E.M.) for the mixture, which produced a 50% anti-proliferative effect in the malignant melanoma cell lines. * *p* < 0.05 vs. the respective IC_50add_ value (Student’s *t*-test with Welch correction). Point M is placed slightly above point A′ for the A357 cell line (**A**) indicating additivity between amantadine and MTO in the MTT assay. For the SK-MEL28 cell line, point M is placed between slightly below point A indicating additive interaction between amantadine and MTO (**B**). For the FM55P cell line (**C**), point M is placed below point A, indicating additive interaction for amantadine with MTO in the MTT assay. For the FM55M2 cell line (**D**), point M is placed significantly below point A (* *p* < 0.05), illustrating synergistic interaction between amantadine and MTO in the MTT assay.

**Figure 11 ijms-23-07653-f011:**
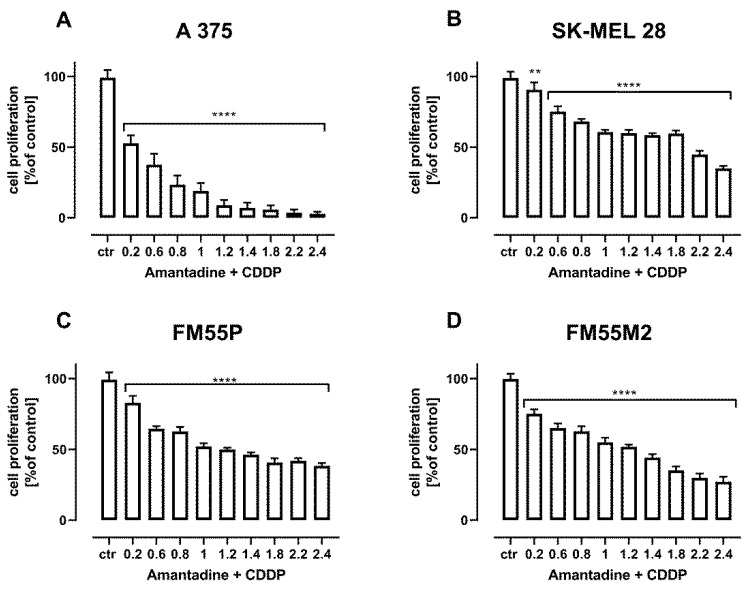
The effect of combination of amantadine and CDDP in fixed ratio of 1:1 on the proliferation of malignant melanoma cell lines (A375 (**A**), SK-MEL28 (**B**), FM55P (**C**) and FM55M2 (**D**)) was measured by BrdU assay after 72 h. Results are presented as mean ± SEM at each concentration. (** *p* < 0.01; **** *p* < 0.0001). The values on axis X represent the multiplicity of calculated IC_50_ (1.0 means half IC_50_ of amantadine + half IC_50_ of CDDP).

**Figure 12 ijms-23-07653-f012:**
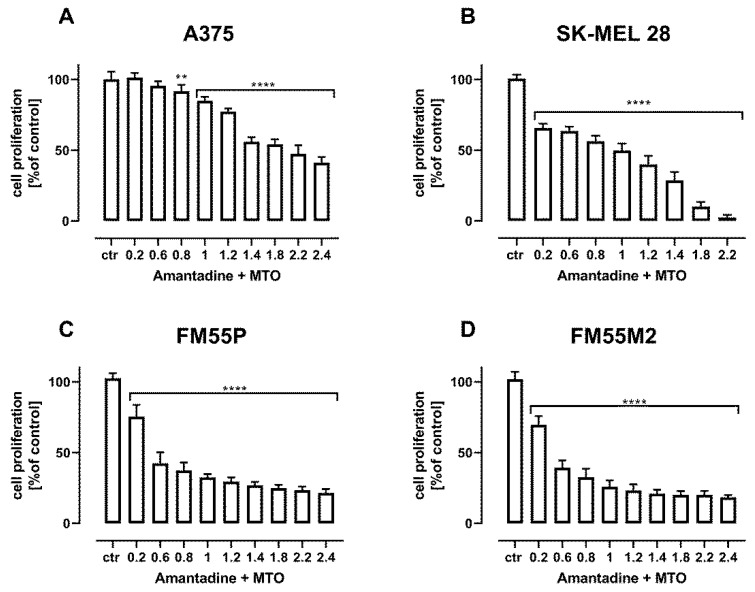
The effect of combination of amantadine and MTO in fixed ratio 1:1 on the proliferation of malignant melanoma cell lines (A375 (**A**), SK-MEL28 (**B**), FM55P (**C**) and FM55M2 (**D**)) was measured by BrdU assay after 72 h. Results are presented as mean ± SEM at each concentration. (** *p* < 0.01; **** *p* < 0.0001). The values on axis X represent the multiplicity of calculated IC_50_ (1.0 means half IC_50_ of amantadine + half IC_50_ of MTO).

**Table 1 ijms-23-07653-t001:** Anti-viabilitive effects of amantadine, mitoxantrone (MTO) and cisplatin (CDDP) administrated separately in melanoma cell lines (A375, SK-MEL28, FM55P and FM5M2) measured in vitro by means of MTT assay.

Cell Line	Drug	IC_50_ ± S.E.M.
A375	Amantadine	58.03 ± 4.17 µg/mL
	CDDP	0.38 ± 0.07 µg/mL
	MTO	0.016 ± 0.006 µg/mL
SK-MEL28	Amantadine	70.68 ± 10.33 µg/mL
	CDDP	0.99 ± 0.21 µg/mL
	MTO	0.901 ± 0.304 µg/mL
FM55P	Amantadine	40.18 ± 12.26 µg/mL
	CDDP	0.54 ± 0.13 µg/mL
	MTO	0.183 ± 0.058 µg/mL
FM55M2	Amantadine	69.46 ± 18.01 µg/mL
	CDDP	0.44 ± 0.10 µg/mL
	MTO	0.066 ± 0.013 µg/mL

**Table 2 ijms-23-07653-t002:** Type I isobolographic analysis of interactions for non-parallel concentration–response relationship lines between amantadine and CDDP at the fixed drug concentration ratio of 1:1 in two melanoma cell lines A375 and SK-MEL28. Results are median inhibitory concentrations (IC_50_ values in μg/mL ± S.E.M.) for two-drug mixtures, determined either experimentally (IC_50exp_) or theoretically calculated (IC_50add_) from the equations of additivity, blocking proliferation in 50% of tested cells in two melanoma cell lines (A375 and SK-MEL28) measured in vitro by the MTT assay. L-IC_50_—lower additive IC_50_ value, U-IC_50_—upper additive IC_50_ value.

Cell Line	IC_50exp_ (μg/mL)	n_exp_	L-IC_50add_ (μg/mL)	n_add_	U-IC_50add_ (μg/mL)	Interaction
A375	7.076 ± 2.002	96	13.901 ± 3.149	286	44.373 ± 3.666	Additivity
SK-MEL28	39.201 ± 8.444	96	30.664 ± 8.861	182	41.165 ± 9.813	Additivity

**Table 3 ijms-23-07653-t003:** Type I isobolographic analysis of interactions for parallel concentration–response relationship lines between amantadine and CDDP at the fixed drug concentration ratio of 1:1 in two melanoma cell lines FM55P and FM55M2. Results are median inhibitory concentrations (IC_50_ values in μg/mL ± S.E.M.) for two-drug mixtures, determined either experimentally (IC_50exp_) or theoretically calculated (IC_50add_) from the equations of additivity, blocking proliferation in 50% of tested cells in two melanoma cell lines (FM55M2 and FM55P) measured in vitro by the MTT assay.

Cell Line	IC_50exp_ (μg/mL)	n_exp_	IC_50add_ (μg/mL)	n_add_	Interaction
FM55M2	41.67 ± 6.68	96	34.95 ± 9.06	164	Additivity
FM55P	23.61 ± 6.22	96	20.36 ± 6.19	164	Additivity

**Table 4 ijms-23-07653-t004:** Type I isobolographic analysis of interactions for non-parallel concentration–response relationship lines between amantadine and MTO at the fixed drug concentration ratio of 1:1 in A375 cell line. Results are median inhibitory concentrations (IC_50_ values in μg/mL ± S.E.M.) for two-drug mixtures, determined either experimentally (IC_50exp_) or theoretically calculated (IC_50add_) from the equations of additivity, blocking proliferation in 50% of tested cells in melanoma cell line A375 measured in vitro by the MTT assay. L-IC_50_—lower additive IC_50_ value, U-IC_50_—upper additive IC_50_ value.

Cell Line	IC_50exp_ (μg/mL)	n_exp_	L-IC_50add_ (μg/mL)	n_add_	U-IC_50add_ (μg/mL)	Interaction
A375	48.19 ± 4.09	96	11.71 ± 5.04	236	46.23 ± 5.25	Additivity

**Table 5 ijms-23-07653-t005:** Type I isobolographic analysis of interactions for parallel concentration-response relationship lines between amantadine and MTO at the fixed drug concentration ratio of 1:1 in three melanoma cell lines, FM55P, FM55M2 and SK-MEL28. Results are median inhibitory concentrations (IC_50_ values in μg/mL ± S.E.M.) for two-drug mixtures, determined either experimentally (IC_50exp_) or theoretically calculated (IC_50add_) from the equations of additivity, blocking proliferation in 50% of tested cells in three melanoma cell lines (FM55M2, FM55P and SK-MEL28) measured in vitro by the MTT assay. * *p* < 0.05 vs. the respective IC_50add_ value.

Cell Line	IC_50exp_ (μg/mL)	n_exp_	IC_50add_ (μg/mL)	n_add_	Interaction
FM55M2	14.39 ± 2.97	96	34.76 ± 9.01 *	164	Synergy
FM55P	9.94 ± 1.96	96	20.18 ± 6.16	164	Additivity
SK-MEL28	33.83 ± 5.80	96	35.79 ± 5.16	164	Additivity

## Data Availability

All data are presented in tabular and graphical forms. Additionally, the data could be available upon request.

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
