# Peer review of "Anticancer Activity of Amantadine and Evaluation of Its Interactions with Selected Cytostatics in Relation to Human Melanoma Cells"

_ijms, 2022, doi:10.3390/ijms23147653_

Round 1

Reviewer 1 Report

The manuscript “Anticancer activity of amantadine and evaluation of its interactions with selected cytostatics in relation to human melanoma cells” documented the antitumor effects of amantadine in combination with cisplatin or mitoxantrone on different melanoma cell lines. The Authors propose to include amantadine in a combined therapy for melanoma with the aim to reduce the dosage of conventional drugs.

In my opinion, the Authors did not investigate the molecular processes underlying the amantadine antitumor activity. Considering that IJMS journal focuses on molecular mechanisms, this manuscript must be implemented before being ready for publication on IJMS.

Major criticisms:

 1.     I appreciated that in the Introduction several information about amantadine mechanism of action in both normal and malignant cells have been added.

Because all cell lines used are BRAF-mutated (this important information must be reported in Introduction or in Materials and Methods Sections), it must be explained why the Authors chose CDDP and MTO instead of the BRAF inhibitors (vemurafenib or dabrafenib). In addition, is not clear why in Parkinson’s patients the melanoma incidence is higher, but the prognosis is better (lines: 46-50). Are there any information included in the epidemiological studies about the therapy assumed by Parkinson’s patients with melanoma? Were the patients treated with amantadine or with other drugs? These points must be better discussed.

2.     I appreciated that the Authors reported new Br-dU data concerning the amantadine antiproliferative effect. However, Br-dU data with combined treatments (amantadine-CDDP and amantadine-MTO) are missing. It would be important to discern whether the additive/synergistic effect of the combined treatments can involve the rate of cell growth, as well.

3.     In many parts of the manuscript (lines: 110; 172; 190; 196; 206; 212; 220; 234; 240; 246; 282; 285; 294 and other) the results of the MTT assay are described as regarding the inhibition of cell proliferation, but this is not correct. MTT assay measures mitochondrial dehydrogenase activity, i.e., all mitochondrial activity of viable cells during the time of incubation of the tetrazolium salts. Thus, the results of this assay, differently to the Br-dU assay, is a picture of cell viability and must be ascribed to the balance between cell growth and cell death. Please, replace in all places “cell proliferation” with “cell viability”.

4.     Flow cytometric analysis data of caspase 3 activity must be implemented by data derived from combined treatments (amantadine-CDDP and amantadine-MTO). With the new data the authors could discern if apoptosis, as well, may be potentiated by a co-treatment.

5.     Even if the authors declare that “the main limitation of this study is the lack of determination of molecular mechanisms of action of drugs in combination” this remains an important critical point of the manuscript. In my opinion, immunoblot analysis must be performed to find differences in the expression level of at least some key proteins. For instance, because apoptosis is involved, Bcl2, Bax, or Apaf-1 expression level could be affected by amantadine. As regard the antiproliferative effect, Cyclins, pRB, pP53, pCDKs, or other cell-cycle regulated proteins, such as the CDK inhibitors: P21, P27, P16, could be targets of amantadine action. The measure of differences in the expression level of some molecular markers could highlight the pathways affected by the claimed pro-apoptotic or antiproliferative activity of amantadine.

6.     To better compare the concentrations between different drugs, drug concentrations must be expressed in μM, instead of μg/ml, because the μM unit of measure is independent to the dissimilar molecular weight of the compounds.

7.     English language needs to be much improved

 Minor criticisms:

1.     replace “melanoma malignant” with “malignant melanoma”.

2.     The meaning of L-IC50add and U-IC50add must be added (lower additive IC50 value and upper additive IC50 value) in the legend of Table 2 and 4.

Author Response

Reviewer_1

The manuscript “Anticancer activity of amantadine and evaluation of its interactions with selected cytostatics in relation to human melanoma cells” documented the antitumor effects of amantadine in combination with cisplatin or mitoxantrone on different melanoma cell lines. The Authors propose to include amantadine in a combined therapy for melanoma with the aim to reduce the dosage of conventional drugs.

In my opinion, the Authors did not investigate the molecular processes underlying the amantadine antitumor activity. Considering that IJMS journal focuses on molecular mechanisms, this manuscript must be implemented before being ready for publication on IJMS.

Major criticisms:

  1. I appreciated that in the Introduction several information about amantadine mechanism of action in both normal and malignant cells have been added.

Because all cell lines used are BRAF-mutated (this important information must be reported in Introduction or in Materials and Methods Sections), it must be explained why the Authors chose CDDP and MTO instead of the BRAF inhibitors (vemurafenib or dabrafenib). In addition, is not clear why in Parkinson’s patients the melanoma incidence is higher, but the prognosis is better (lines: 46-50). Are there any information included in the epidemiological studies about the therapy assumed by Parkinson’s patients with melanoma? Were the patients treated with amantadine or with other drugs? These points must be better discussed.

  1. I appreciated that the Authors reported new Br-dU data concerning the amantadine antiproliferative effect. However, Br-dU data with combined treatments (amantadine-CDDP and amantadine-MTO) are missing. It would be important to discern whether the additive/synergistic effect of the combined treatments can involve the rate of cell growth, as well.

Reply:

We have performed additional experiments illustrating the antiproliferative effects of the combination of amantadine with CDDP and MTO in BrdU test (new figures 11 and 12) as suggested.

  1. In many parts of the manuscript (lines: 110; 172; 190; 196; 206; 212; 220; 234; 240; 246; 282; 285; 294 and other) the results of the MTT assay are described as regarding the inhibition of cell proliferation, but this is not correct. MTT assay measures mitochondrial dehydrogenase activity, i.e., all mitochondrial activity of viable cells during the time of incubation of the tetrazolium salts. Thus, the results of this assay, differently to the Br-dU assay, is a picture of cell viability and must be ascribed to the balance between cell growth and cell death. Please, replace in all places “cell proliferation” with “cell viability”.

Reply:

We have replaced the term “cell proliferation” with “cell viability” throughout the manuscript as recommended.

  1. Flow cytometric analysis data of caspase 3 activity must be implemented by data derived from combined treatments (amantadine-CDDP and amantadine-MTO). With the new data the authors could discern if apoptosis, as well, may be potentiated by a co-treatment.

Reply:

We have added the graphs illustrating the effects of the combination of amantadine with CDDP and MTO, as suggested (Figure 4)

  1. Even if the authors declare that “the main limitation of this study is the lack of determination of molecular mechanisms of action of drugs in combination” this remains an important critical point of the manuscript. In my opinion, immunoblot analysis must be performed to find differences in the expression level of at least some key proteins. For instance, because apoptosis is involved, Bcl2, Bax, or Apaf-1 expression level could be affected by amantadine. As regard the antiproliferative effect, Cyclins, pRB, pP53, pCDKs, or other cell-cycle regulated proteins, such as the CDK inhibitors: P21, P27, P16, could be targets of amantadine action. The measure of differences in the expression level of some molecular markers could highlight the pathways affected by the claimed pro-apoptotic or antiproliferative activity of amantadine.

Reply:

We have performed additional experiments with western blot analysis for amantadine in various cell lines, as recommended (Figure 6).

  1. To better compare the concentrations between different drugs, drug concentrations must be expressed in μM, instead of μg/ml, because the μM unit of measure is independent to the dissimilar molecular weight of the compounds.

Reply:

We have added the range concentrations of the used drugs expressed in μM (page 9).

  1. English language needs to be much improved

 Minor criticisms:

  1. replace “melanoma malignant” with “malignant melanoma”.

Reply:

We have changed the term “malignant melanoma” accordingly.

  1. The meaning of L-IC50add and U-IC50add must be added (lower additive IC50 value and upper additive IC50 value) in the legend of Table 2 and 4.

Reply:

We have explained the used abbreviations in the legend, as recommended.

Reviewer 2 Report

The paper is interesting, however, needs some improvement:

1. Figure 1 should be divided into separated graphs, by adding A), B), C) ... - by doing so, the authors may further easily refer to the specific part of the data. 

2. Text size of all graphs in each figures should be the same - not acceptable to have smaller size "HaCaT" like in figure 1. 

3. HaCaT viability was the first affected by amantadine, but the authors claim: "keratinocyte cell line (HaCaT) was only slightly inhibited, 106 which confirms the specific effect of amantadine on melanoma cells" this should be clarified in text. We can also see that both "normal" cell lines were the ones to be affected by really low concentration of amantadine.

4. Please clarify which doses of the drugs used for the study may be found in human body after their administration.

5. Why you have reduced the number of cell lines in figure 2?

6. Control sample in figure 3 should be named differently because it is very confusing - your ctr+ is in fact a negative control and ctr is a positive control for the experiment. 

7. Maybe adding bar plots for figure 4 and 5 would help the reader to understand your results at first glance on the paper?

8. Why you used the fixed ratio of 1:1 between amantadine and MTO or CDDP? Do they occur in the human body in the same concentration? Wouldn't it be better to use the IC50 concentration of the antimelanotic agent and a real world concentration of amantadine? In human body amantadine probably do not tend to have a standalone antimelanotic activity...

9. I don't get the idea of using such a theoretical analysis of interaction between drugs when you can perform the experiments in vitro with two of the drugs added simultaneously to the cells and compare it with the standalone drugs. Maybe after the analysis, an in vitro validation of the theoretical data would be a good idea?

Otherwise an interesting paper.

Author Response

Reviewer_2

The paper is interesting, however, needs some improvement:

  1. Figure 1 should be divided into separated graphs, by adding A), B), C) ... - by doing so, the authors may further easily refer to the specific part of the data. 

Reply:

We have prepared a multipart figure (A-F) as suggested.

  1. Text size of all graphs in each figures should be the same - not acceptable to have smaller size "HaCaT" like in figure 1. 

Reply:

We have corrected the size of letters.

  1. HaCaT viability was the first affected by amantadine, but the authors claim: "keratinocyte cell line (HaCaT) was only slightly inhibited, 106 which confirms the specific effect of amantadine on melanoma cells" this should be clarified in text. We can also see that both "normal" cell lines were the ones to be affected by really low concentration of amantadine.
  2. Please clarify which doses of the drugs used for the study may be found in human body after their administration.

Reply:

This in vitro study is a first step in cancer (preclinical) research. It is too early to define the doses (concentrations ) of the drugs observed in human, especially for amantadine. Noteworthy, CDDP and MTO are used clinically as the antimelanotic drugs in the second line treatment in metastatic melanoma.

  1. Why you have reduced the number of cell lines in figure 2?
  2. Control sample in figure 3 should be named differently because it is very confusing - your ctr+ is in fact a negative control and ctr is a positive control for the experiment. 

Reply:

We have changed the name of the controls as suggested.

  1. Maybe adding bar plots for figure 4 and 5 would help the reader to understand your results at first glance on the paper?

Reply:

Figures 4 and 5 were originally created by the cytometer and it is a good practice to present the originally created figures instead of modified graphs. The same holds true for western blot graphs.

  1. Why you used the fixed ratio of 1:1 between amantadine and MTO or CDDP? Do they occur in the human body in the same concentration? Wouldn't it be better to use the IC50 concentration of the antimelanotic agent and a real world concentration of amantadine? In human body amantadine probably do not tend to have a standalone antimelanotic activity...

Reply:

Isobolography is based on concentrations of drugs fully effective in this experiment i.e., both amantadine and CDDP or MTO should exert the antimelanocytic effects in various melanoma cell lines. This is the reason that amantadine was used at its IC50 dose in this study.

  1. I don't get the idea of using such a theoretical analysis of interaction between drugs when you can perform the experiments in vitro with two of the drugs added simultaneously to the cells and compare it with the standalone drugs. Maybe after the analysis, an in vitro validation of the theoretical data would be a good idea?

Reply:

Theoretically calculated additive concentrations (IC50 add values) were always experimentally verified with the experimentally derived concentrations (IC50 exp values) in in vitro studies. Isobolographical analysis is not a theoretical analysis. Concentrations accepted as additive (that are theoretically calculated from the Loewe equation) are always statistically verified with their experimentally derived values providing finally the types of interaction between the tested drugs.

Otherwise an interesting paper.

Round 2

Reviewer 1 Report

I appreciated the authors' help in implementing their manuscript.

It seems to me that at least the new results for the analysis of apoptosis with the combination of drugs are very interesting. It’s a shame they’re not reported in the abstract and discussed in Discussion Section.

Minor Criticisms:

Replace "proliferative" with "viability” in the multiple parts of the Discussion Section.

It is not clear whether the results, however interesting, of Immunoblots are significant or not because there is no quantitative analysis. Add the quantitative analysis.

Add a hint of significant new results in the Abstract

Implement the Discussion Section by discussing the new significant results.

In my opinion, after these adjustments the work can be published on IJMS

Author Response

Minor Criticisms:

Replace "proliferative" with "viability” in the multiple parts of the Discussion Section.

Reply: We have changed this term accordingly.

It is not clear whether the results, however interesting, of Immunoblots are significant or not because there is no quantitative analysis. Add the quantitative analysis.

Reply: The quantitative analysis of immunoblots has been added as multi-part graphs to the Figure 6, as suggested.

Add a hint of significant new results in the Abstract

Reply: Such information has been added.

Implement the Discussion Section by discussing the new significant results.

Reply: Discussion has been changed following the Reviewer’ suggestion.

This manuscript is a resubmission of an earlier submission. The following is a list of the peer review reports and author responses from that submission.

Round 1

Reviewer 1 Report

The authors submitted an original article, in which the effect of amantadine and its interactions with cytostatics in melanoma cells have been investigated. The experiments presented are well designed, and the findings of this paper are interesting. The presented results show significant novel aspects that could be also potentially translated to the clinical care of melanoma patients in the future. However, there are several issues to be addressed by the authors. In conclusion, this manuscript should undergo a major revision to be deemed for publication.

Major issues:

  1. It should be emphasized that the first-line treatment of metastatic melanoma is target therapy or immune-therapy. The investigated interactions of amantadine with cytostatics are only relevant for second-line treatment with agents such as cisplatin.
  2. To improve the manuscript, the introduction would need additional data to put more emphasis on the description of current knowledge on amantadine instead of describing survival rates for early melanoma and a long paragraph on cisplatin and mitoxantrone.
  3. Would the authors consider the use of amantadine on further cell lines, not only melanoma and keratinocyte cell line to reveal if it has a selective effect on melanoma cells.
  4. It should be addressed why the effect of amantadine differs greatly on different melanoma cell lines.
  5. In the Discussion, the part on the association of Parkinson’s disease and melanoma is too long and out of focus.

Minor issues:

  1. To discuss the clinical implications of the results, survival rates of metastatic melanoma patients with cytostatic treatment should be described.
  2. Mitoxantrone should not be abbreviated as “MTX”, as the readers might assume methotrexate. Please use the abbreviation “MTO”, that is more commonly used.
  3. In the “Concentration-response relationship curves” it is difficult to read the y and R values on the graphs, please place these to the figure legends.
  4. For the legend of Figure 5, please clarify the description, as it is hard to comprehend the findings of the graphs.
  5. The English of the manuscript needs major improvements.

Reviewer 2 Report

Authors have to change the order of the chapters: 1. Introduction 2. Materials and methods 3. Results 4. Discussion 5. Conclusion.

They also need to revise the abstract to make it more comprehensible and complete with respect to the article content.

Reviewer 3 Report

In the manuscript “Anticancer activity of amantadine and evaluation of its interactions with selected cytostatics in relation to human melanoma cells” Krasowska et al. investigate the effect of amantadine, a common antiparkinson drug, in combination with cisplatin or mitoxantrone on different melanoma cell lines. The manuscript is interesting, considering that melanoma correlates with a high mortality rate as no effective treatments are currently available in clinic. Therefore, in this context, finding alternative therapeutical strategies is crucial. However, in my opinion, the results reported in this paper are too preliminary: more experiments must be added. The authors did not verify whether amantadine effect is cytostatic, cytotoxic or both. Moreover, considering that the IJMS journal focuses on molecular sciences, this manuscript results to be not appropriate for publication as molecular mechanisms are not investigated at all.

Major criticisms:

  1. In the introduction more information on amantadine mechanism of action should be added. Moreover, drawbacks related to cisplatin and mitoxantrone treatments in melanoma should be mentioned.
  2. HaCaT cells were used to demonstrate that amantadine treatment is melanoma specific. Considering that melanoma derives from the neoplastic transformation of melanocytes, why not using normal human epidermal melanocytes (NHEM) instead of keratinocytes?
  3. In the results section the authors should report the histograms regarding the anti-proliferative effect of the combination treatments.
  4. All the results in the paper are presented as mean ± standard error (S.E.M.) but it is more appropriate to present them as mean ± standard deviation (S.D.).
  5. In Figure 2 statistical analysis must be reported. Moreover, amantadine treatment seems to have no significant effect on A375 and SK-MEL28 cell lines.
  6. Line 14: the authors claim that amantadine has an “antiproliferative potential” but no experiments confirmed this statement. Bromouridine incorporation assay, should be performed to evaluate amantadine antiproliferative effect and whether the combined treatment may improve such ability.
  7. Lines 230-232: No experimental data and/or bibliographic citations are reported in the paper to support the sentence “amantadine is not a cytostatic drug”. LDH assay is used as cytotoxicity assessment, but it is well known that this kind of test evaluates necrosis. Moreover, a good anticancer drug should trigger apoptosis, ferroptosis or autophagy rather than necrosis. In fact, necrosis correlates with increased inflammation that may lead to tissue damage and therefore be considered a side effect.

My suggestion is to perform cytofluorimetric analysis (Annexin V/propidium iodide) to understand whether treated cells undergo apoptosis and, in that case, immunoblot analysis may be used to confirm an altered expression level of proteins considered to be apoptotic markers.

  1. Lines 235-237: considering the histogram in Figure 1 amantadine slightly affects keratinocytes as well.
  2. Lines 238-239: even if the authors declare that “the main limitation of this study is the lack of determination of molecular mechanisms of action of drugs in combination” immunoblot analysis must be performed at least to understand the differences in protein expression levels between control and amantadine treated samples. These experiments may highlight the molecular mechanism responsible for the claimed antiproliferative effect of amantadine.

Minor criticisms:

  1. Line 17, line 204: replace “melanoma malignant” with “malignant melanoma”.
  2. Line 45-46: explain the sentence: “due to high resistance it has emerged”.
  3. Line 218: replace “resistible” with “resistant”.
  4. Express treatment concentrations in μM instead of μg/ml.

Despite the promising adjuvant effect of amantadine over other chemotherapeutic agents commonly used in clinic to counteract melanoma this work is too preliminary. I suggest resubmitting the manuscript only after the results implementation.